# Assessment of CNN-Based Methods for Individual Tree Detection on Images Captured by RGB Cameras Attached to UAVs

**DOI:** 10.3390/s19163595

**Published:** 2019-08-18

**Authors:** Anderson Aparecido dos Santos, José Marcato Junior, Márcio Santos Araújo, David Robledo Di Martini, Everton Castelão Tetila, Henrique Lopes Siqueira, Camila Aoki, Anette Eltner, Edson Takashi Matsubara, Hemerson Pistori, Raul Queiroz Feitosa, Veraldo Liesenberg, Wesley Nunes Gonçalves

**Affiliations:** 1Faculty of Computer Science, Federal University of Mato Grosso do Sul, Campo Grande 79070-900, Brazil; 2Faculty of Engineering, Architecture and Urbanism and Geography, Federal University of Mato Grosso do Sul, Campo Grande 79070-900, Brazil; 3Department of Computer Engineering, Dom Bosco Catholic University, Campo Grande 79117-900, Brazil; 4CPAQ, Federal University of Mato Grosso do Sul, Aquidauana 79200-000, Brazil; 5Institute of Photogrammetry and Remote Sensing, Technische Universität Dresden, 01062 Dresden, Germany; 6Department of Electrical Engineering, Pontifical Catholic University of Rio de Janeiro, Rio de Janeiro 22451-900, Brazil; 7Department of Forest Engineering, Santa Catarina State University, Lages 88520-000, Brazil

**Keywords:** object-detection, deep learning, remote sensing

## Abstract

Detection and classification of tree species from remote sensing data were performed using mainly multispectral and hyperspectral images and Light Detection And Ranging (LiDAR) data. Despite the comparatively lower cost and higher spatial resolution, few studies focused on images captured by Red-Green-Blue (RGB) sensors. Besides, the recent years have witnessed an impressive progress of deep learning methods for object detection. Motivated by this scenario, we proposed and evaluated the usage of Convolutional Neural Network (CNN)-based methods combined with Unmanned Aerial Vehicle (UAV) high spatial resolution RGB imagery for the detection of law protected tree species. Three state-of-the-art object detection methods were evaluated: Faster Region-based Convolutional Neural Network (Faster R-CNN), YOLOv3 and RetinaNet. A dataset was built to assess the selected methods, comprising 392 RBG images captured from August 2018 to February 2019, over a forested urban area in midwest Brazil. The target object is an important tree species threatened by extinction known as *Dipteryx alata* Vogel (Fabaceae). The experimental analysis delivered average precision around 92% with an associated processing times below 30 miliseconds.

## 1. Introduction

Preservation of sensitive tree species requires timely and accurate information on their distribution in the area under threat. Remote sensing techniques have been increasingly applied as alternatives to costly and time consuming field surveys for assessing forest resources. For this purpose, satellite, aerial and, more recently, Unmanned Aerial Vehicle (UAV) have been the most common platforms used for data collection.

Multispectral [1,2,3,4,5] and hyperspectral [6,7] imageries, Light Detection And Ranging (LiDAR) data [8,9,10,11], and also combinations of them [12,13,14,15,16,17] have been the preferred data source. Clark et al. [6] used airborne hyperspectral data (161 bands, 437–2434 nm) for the classification of seven tree species. Linear discriminant analysis (LDA), maximum likelihood (ML) and spectral angle mapper (SAM) classifiers were tested. The authors reported accuracy of 88% with a ML classifier based on 60 bands. Dalponte et al. [7] investigated the use of hyperspectral sensors for the classification of tree species in a boreal forest. Accuracy around 80% was achieved, using Support Vector Machines (SVM) and Random Forest (RF) classifiers.

Immitzer et al. [3] applied RF to classify 10 tree species in an Austrian temperate forest upon WorldView-2 (8 spectral bands) multispectral data, having achieved an overall classification accuracy around 82%. In a later work, Immitzer et al. [4] used Sentinel-2 multispectral imagery to classify tree species in Germany with a RF classifier achieving accuracy around 65%. Franklin and Ahmed [5] reported 78% accuracy in the classification of deciduous tree species applying object-based and machine learning techniques to UAV multispectral images.

Voss and Sugumaran [12] combined hyperspectral and LiDAR data to classify tree species using an object-oriented approach. Accuracy improvements up to 19% were achieved when both data were combined. Dalponte et al. [15] investigated the combination of hyperspectral and multispectral images with LiDAR for the classification of tree species in Southern Alps. They achieved 76.5% accuracy in experiments using RF and SVM. Nevalainen et al. [18] combined UAV-based photogrammetric point clouds and hyperspectral data for tree detection and classification in boreal forests. RF and Multilayer Perceptron (MLP) provided 95% overall accuracy. Berveglieri et al. [19] developed a method based on multi-temporal Digital Surface Model (DSM) and superpixels for classifying successional stages and their evolution in tropical forest remnants in Brazil.

While numerous studies have been conducted on multispectral, hyperspectral, LiDAR and combinations of them, there are few studies relying on RGB images for tree detection/classification. Feng et al. [20] used RGB images for urban vegetation mapping. They used RF classifiers, and verified that the texture, derived from the RGB images, contributed significantly to improve the classification accuracy. However, tree species classification was not specifically addressed in this work.

In the last few years, approaches based on deep learning, such as Convolutional Neural Networks (CNNs) and their variants, gained popularity in many fields, including remote sensing data analysis. Mizoguchi et al. [11] applied CNN to terrestrial LiDAR data to classify tree species and achieved between 85% and 90% accuracy. Weinstein et al. [21] used semi-supervised deep learning neural networks for individual tree-crown detection in RGB airborne imagery. Barré et al. [22] developed a deep learning system for classifying plant species based on leaf images using CNN.

Regarding plant species classification and diseases detection based on leaf images, several works were developed [23,24,25,26,27,28]. Fuentes et al. [25] focused on the development of a deep-learning-based detector for real-time tomato plant diseases and pests recognition, considering three CNNs: Faster Region-based Convolutional Neural Network (Faster R-CNN), Region-based Fully Convolutional Network (R-FCN) and Single Shot Multibox Detector (SSD). However, tree detection was not the target application.

To the best of our knowledge, no study focused thus far on state-of-the-art deep learning-based methods for tree detection on images generated by RGB cameras on board of UAVs. The present study addressed this gap and presented an evaluation of deep learning-based methods for individual tree detection on UAV/RGB high resolution imagery. This study focused on a tree species known as *Dipteryx alata* Vogel (Fabaceae), popularly known as baru or cumbaru (henceforth cumbaru), which is threatened by extinction according to the IUCN (2019) (The International Union for Conservation of Nature’s Red List of Threatened Species, https://www.iucnredlist.org/species/32984/9741012). This species has a high economic potential and a wide range of applications, mainly for the use of non-timber forest products. It is distributed over a large territory, being mostly associated to the Brazilian Savanna, although it also occurs in the wetlands [29] in midwest Brazil.

Our work hypothesis is that state-of-the-art deep learning-based methods are able to detect single tree species upon high-resolution RGB images with attractive cost, accuracy and computational load. The contribution of this work is twofold. First, we assessed the usage of high-resolution images produced by RGB cameras carried by UAVs for individual trees detection. Second, we compared three state-of-the-art CNN-based object detection methods, namely FasterRCNN, RetinaNet and YOLOv3, for the detection of cumbaru trees on said UAV/RGB imagery.

The rest of this paper is organized as follows. Section 2 presents the materials and methods adopted in this study. Section 3 presents and discusses the results obtained in the experimental analysis. Finally, Section 4 summarizes the main conclusions and points to future directions.

## 2. Materials and Methods

### 2.1. Overall Experimental Procedure

The experiments were conducted in four main steps (see Figure 1). First, the images were acquired in different periods of the year by a RGB camera on a UAV platform. The images were annotated by a specialist who delimited the cumbaru trees with a rectangle (bounding box). In the next step, the networks representing each method were trained to detect the cumbaru tree instances in a set of training images. In the third step, the images not used for training were submitted to the trained networks, which predicted the cumbaru tree occurrences, returning the detected bounding boxes. In the final step, the accuracy metrics were computed for each methods on the predicted results.

### 2.2. Data Acquisition

In total, 392 UAV images were acquired over seven months (from August 2018 to February 2019 in six missions). An advanced Phantom 4 UAV equipped with a 20-megapixel camera captured the images at flight heights of 20–40 m (see Table 1 for more details). Images with a mean Ground Sample Distance (GSD) of 0.82 cm (centimeter) were acquired in three study regions, with a total area of approximately 150,000.00 square meters, in the urban part of Campo Grande municipality, in the Brazilian state of Mato Grosso do Sul. Approximately 110 trees were imaged during the missions. Some tree samples are shown in Figure 2. Notice the variability in terms of appearance, scale and illumination.

Each image was annotated by a specialist using LabelMe software (https://github.com/wkentaro/labelme). In this process, a bounding box specified by the top-left and bottom-right corners was annotated for each cumbaru tree sample in the image.

### 2.3. Object Detection Methods

The object detection methods compared in this study are briefly described in the following (the following source codes were used as a basis for our implementation: Faster-RCNN, https://github.com/yhenon/keras-frcnn; YOLOv3, https://github.com/qqwweee/keras-yolo3; and RetinaNet, https://github.com/fizyr/keras-retinanet).

Faster-RCNN [30]: In this method, a feature map is initially produced by a ResNet50 [31]. Given the feature map, Faster-RCNN detects object instances in two stages. The first stage, called Region Proposal Network (RPN), receives the feature map and proposes candidate object bounding boxes. The second stage also accesses the feature map and extracts features from each candidate bounding box using a Region of Interest Pooling (RoIPoolRoIPool) layer. This operation is based on max pooling, and aims to obtain a fixed-size feature map, independent on the size of the candidate bounding box at its input. A softmax layer then predicts the class of the proposed regions as well as the offset values for their bounding boxes.YOLOv3 [32]: Unlike Faster-RCNN, which has a stage for region proposal, YOLOv3 addresses the object detection as a problem of direct regression from pixels to bounding box coordinates and class probabilities. The input image is divided into S×S tiles. For each tile, YOLOv3 predicts bounding boxes using dimension clusters as anchor boxes [33]. For each bounding box, an objectness score is predicted using logistic regression, which indicates the chance of the bounding box to have an object of interest. In addition, *C* class probabilities are estimated for each bounding box, indicating the classes that it may contain. In our case, each bounding box may contain the cumbaru species or background (uninteresting object). Thus, each prediction in YOLOv3 is composed of four parameters for the bounding box (coordinates), one objectness score and *C* class probabilities. To improve detection precision, YOLOv3 predicts boxes at three different scales using a similar idea to feature pyramid networks [34]. As a backbone, YOLOv3 uses Darknet-53 as it provides high accuracy and requires fewer operations compared to other architectures.RetinaNet [35]: Similar to YOLOv3, RetinaNet is a one-stage object detector but it addresses class imbalance by reducing the loss assigned to well-classified images. Class imbalance occurs when the number of background examples is much larger than examples of the object of interest (cumbaru trees). Using this new loss function, training focuses on hard examples and prevents the large number of background examples from hampering method learning. RetinaNet architecture consists of a backbone and two task-specific subnetworks (see Figure 1b). As the backbone, RetinaNet adopts the Feature Pyramid Network from [34], which is responsible for computing a feature map over an entire input image. The first subnet is responsible for predicting the probability of object’s presence at each spatial position. This subnet is a small Fully Convolutional Network (five conv layers) attached to the backbone. The second subnet, which is parallel with the object classification subnet, performs bounding box regression. The design of this subnet is identical to the first one except that it estimates box coordinates for each spatial location at the end.

### 2.4. Experimental Setup

We adopted in our experiments a five-fold cross validation strategy. Thus, all images were randomly divided into five equal sized sets, called folds. One fold was separated for testing while the remaining four folds were used as training data. This procedure was repeated five times, each time with a different fold selected for testing. Part of the training set was used for validation. Thus, each round (or iteration) of the cross validation procedure was carried out on training, validation, and testing sets comprising 60%, 20% and 20% of the available images, respectively. To reduce the risk of overfitting, we tuned the learning rate and the number of epochs upon the validation set. The weights of the ResNet backbone were initialized with values pre-trained at ImageNet, a procedure known as transfer learning.

The Adam optimizer was used for training all object-detection methods. We set the learning rate to 0.01,0.001,0.0001 and 0.00001. The networks were trained through a number of epochs until the loss stabilized both in the training and in the validation sets. After this tuning procedure, we adopted learning rates equal to 0.0001, 0.001 and 0.0001, and the number of epochs was set to 500, 600, and 250 for Faster-RCNN, YOLOv3, and RetinaNet, respectively.

The performance of each method is reported in the next section by precision–recall curves and the average precision (AP) [36,37]. To estimate the precision and recall, we calculated the Intersection over Union (IoU), which is given by the overlapping area between the predicted and the ground truth bounding boxes divided by the area of union between them. Following well-known competitions in object detection [36,37], a correct detection (True Positive, TP) is considered for IoU ≥ 0.5, and a wrong detection (False Positive, FP) for IoU < 0.5. A False Negative (FN) is assigned when no corresponding ground truth is detected. Given the above metrics, precision and recall are estimated using Equations (Equation 1) and (Equation 2), respectively.

(1)$$P=\frac{TP}{TP+FP}$$

(2)$$R=\frac{TP}{TP+FN}$$

The average precision is given by the area under the precision–recall curve.

## 3. Results and Discussion

This section presents the results collected in our experiments in three ways. Section 3.1 reports the performance quantitatively in terms of average precision. Section 3.2 presents qualitative results. Finally, we discuss in Section 3.3 the computational costs.

### 3.1. Precision Results of Three CNN-Based Methods

Figure 3 presents the precision–recall curves of all tested variants for each cross validation round. RetinaNet delivered consistently the highest precision and recall among all tested methods. Despite the comparatively smaller IoUs, Faster-RCNN and YOLOv3 also achieved encouraging results considering the complexity of the problem, as the dataset contains many similar trees.

The average precision (area under the precision–recall curve) of the detection methods in each cross validation round is shown in Table 2. RetinaNet presented the most accurate results, 92.64% (±2.61%) on average over all five rounds. Actually, RetinaNet consistently achieved the best results on all folds. YOLOv3 and Faster-RCNN came next, with average precision 85.88% (±4.03%) and 82.48% (±3.94%), respectively. In accordance with Figure 3, Table 2 indicates that RetinaNet outperformed its counterparts by about 7%, reaching 92.64% average precision. RetinaNet proposed a new loss function to focus learning on hard negative examples [35]. In this case, training focused on separating the cumbaru from similar trees (hard examples) contributed to greater precision. It is worth emphasizing that the dataset represents a wide variety of environmental conditions, such as flight height and lighting. Thus, the results support the hypothesis that high resolution UAV/RGB images might be a viable approach for detection of individual trees.

### 3.2. Detection under Different Conditions

Figure 4 shows detection results in different seasons, as cumbaru trees have different color and overall appearance. The first row of images shows the cumbaru with chestnuts while the second row presents the cumbaru with greener leaves. The images were captured approximately five months apart from each other. In contrast to other detection approaches (e.g., [12]), all tested methods managed to perform well regardless of image acquisition dates. Previous work suggested periods of the year best suited for capturing images (e.g., September [38] and October [39]). Voss and Sugumaran [12] showed that methods trained in images captured in the fall present more consistent results to those trained with images captured in the summer. On the other hand, the methods used in this work do not need to be trained separately in each season and present consistent precision compared to the literature methods.

The methods were able to detect cumbaru trees even on images captured under different lighting and scale conditions, as shown in Figure 5. The first column shows the ground truth while the three columns on the right present the results produced by the three detection methods.

The UAV flight height directly influences the scale of a tree image. Generally, all tested methods were able to handle different scales and flight heights in the range represented in the dataset. Figure 5 illustrates how they behaved under this kind of variation.

Figure 6 shows results of the same tree captured from different view angles.

### 3.3. Discussion on Computational Complexity

The models were trained and tested on a desktop computer with an Intel(R) Xeon(R) CPU E3-1270@3.80GHz, 64 GB memory, and NVIDIA Titan V graphics card (5120 Compute Unified Device Architecture (CUDA) cores and 12 GB graphics memory). The detection algorithms were coded using Keras-Tensorflow [40] on the Ubuntu 18.04 operating system.

Table 3 shows the mean and standard deviation of the runtime for a tree detection after the image image and trained network model have been loaded.

As expected, the Faster-RCNN variant had the highest computational cost, because it comprises two sequential stages, the first one to propose regions, followed by the second one that classifies the proposed regions. YOLOv3 and RetinaNet were approximately 6.3 and 2.5 times faster than Faster-RCNN, respectively, mainly because they handle object detection as a regression problem.

The results in Table 3 suggest that the methods meet real-time requirements and may be embedded in devices with comparatively low computational capacity.

## 4. Conclusions

In this work, we proposed and evaluated an approach for the detection of tree species based on CNN and high resolution images captured by RGB cameras in an UAV platform. Three state-of-the-art CNN-based methods for object detection were tested: Faster R-CNN, YOLOv3 and RetinaNet. In the experiments carried out on a dataset comprising 392 images, RetinaNet achieved the most accurate results, having delivered 92.64% average precision. Regarding computational cost, YOLOv3 was faster than its counterparts. Faster RCNN was the least accurate and at the same time the most computationally demanding among the assessed detection methods.

The experimental results indicate that RGB cameras attached to UAVs and CNN-based detection algorithms constitute a promising approach towards the development of operational tools for population estimates of tree species, as well for demography monitoring, which is fundamental to integrate economic development and nature conservation. Future works will investigate the application of the proposed techniques considering other tree species. Real-time tree detection using embedded devices will also be investigated.

## Figures and Tables

**Figure 1 sensors-19-03595-f001:**
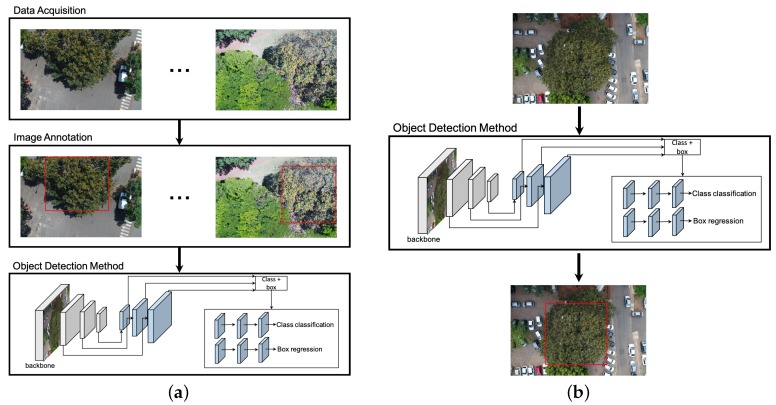
General processing chain: (**a**) UAV images at different seasons were captured and annotated by a specialist. A set of images were selected to train the detection network. (**b**) Once trained, the network was applied to detect cumbaru trees in test images. The object detection method in this figure corresponds to RetinaNet, although other methods (e.g., Faster-RCNN and YOLOv3) can be applied.

**Figure 2 sensors-19-03595-f002:**
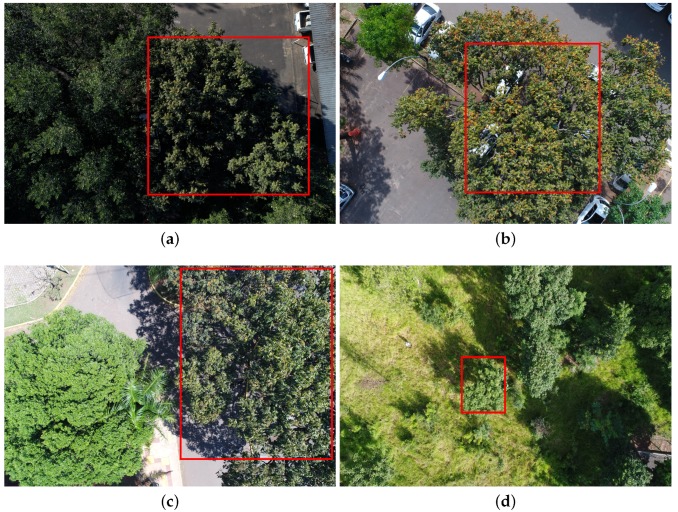
Examples from the dataset: images captured on: (**a**) 26 August 2018; (**b**) 21 September 2018; (**c**) 22 September 2018; and (**d**) 20 February 2019.

**Figure 3 sensors-19-03595-f003:**
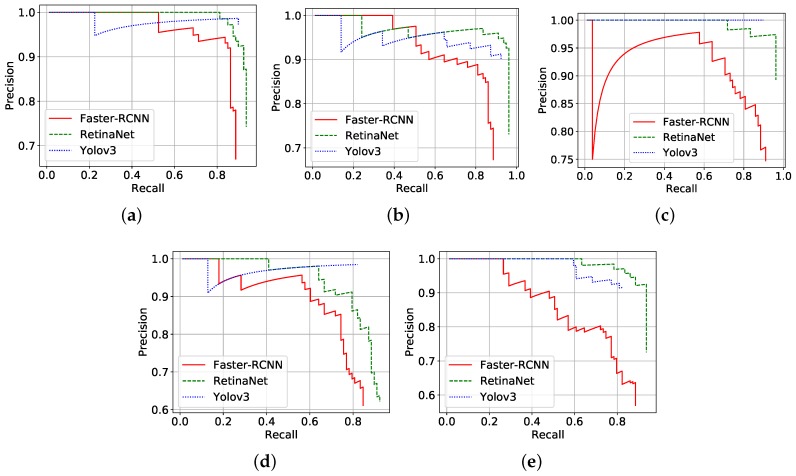
Precision–recall curves of detection methods in all five cross validation rounds (**a**–**e**).

**Figure 4 sensors-19-03595-f004:**
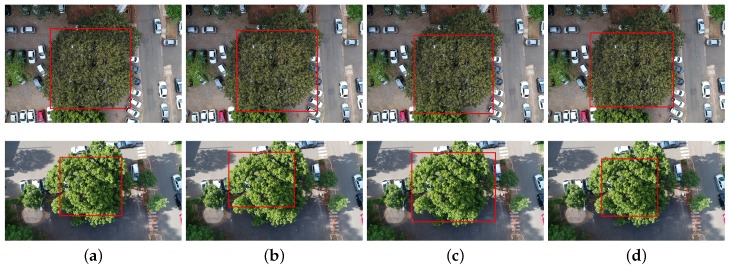
Examples of detection results in images captured in different seasons: (**a**) ground truth; (**b**) Faster-RCNN; (**c**) YOLOv3; and (**d**) RetinaNet.

**Figure 5 sensors-19-03595-f005:**
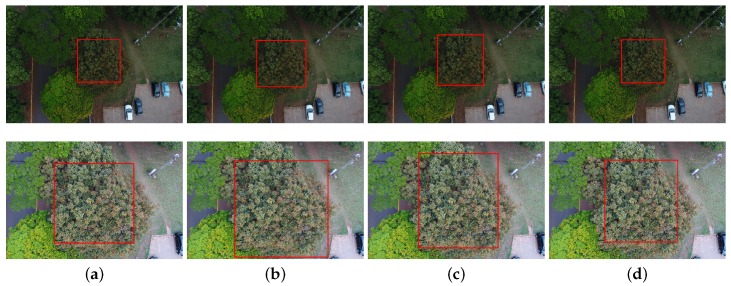
Examples of detection results in images captured for different lighting (average of 67.15 and 130.99 for the brightness channel of the HSB color space) and scale conditions (1:4000 and 1:2500): (**a**) ground truth; (**b**) Faster-RCNN; (**c**) YOLOv3; and (**d**) RetinaNet.

**Figure 6 sensors-19-03595-f006:**
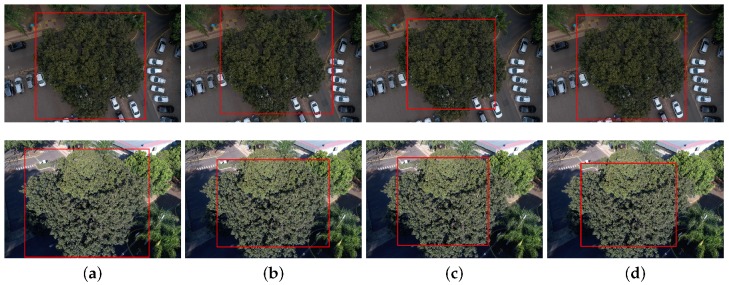
Examples of detection results in images with different capture angles (0° and 30°): (**a**) ground truth; (**b**) Faster-RCNN; (**c**) YOLOv3; and (**d**) RetinaNet.

**Table 1 sensors-19-03595-t001:** Aircraft and flight specifications.

Aircraft	Sensor	Field of View	Nominal Focal Length	Image Size	Mean GSD	Mean Flight Height
Phantom4	1” CMOS	84°	8.8 mm	5472 × 3648	0.82 cm	30 m
Advanced				(20 Mp)		

**Table 2 sensors-19-03595-t002:** Average precision (%) for cumbaru tree detection in five cross validation rounds (R1–R5).

Variant	R1	R2	R3	R4	R5	Mean (std)
Faster-RCNN	86.62	84.14	86.13	77.83	77.69	82.48 (±3.94)
YOLOv3	89.08	88.64	89.74	80.99	80.93	85.88 (±4.03)
RetinaNet	93.13	93.92	95.65	87.82	92.66	92.64 (±2.61)

**Table 3 sensors-19-03595-t003:** Computational cost of the proposed approach variants. The time is the average in seconds to execute the deep learning methods on an image.

Approach Variation	Time (s)
Faster-RCNN	0.163 (±0.066)
YOLOv3	0.026 (±0.001)
RetinaNet	0.067 (±0.001)

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
