# Peer review of "Assessment of CNN-Based Methods for Individual Tree Detection on Images Captured by RGB Cameras Attached to UAVs"

_sensors, 2019, doi:10.3390/s19163595_

Round 1

Reviewer 1 Report

By using RGB images taken from an off-the-shelf UAV and three CNN-based object detection methods (Faster R-CNN, YOLOv3 and RetinaNet), this study was able to identify a kind of tree species under threat: Fabaceae. The overall results are relatively good with the average precision range from 82.48% to 92.64%.

Introduction part should be improved to combine section 2. The discussion is not enough and I suggest that the authors should separate it from results.

Please find the detailed suggestion as follows:

1.    Introduction:

Line 17: Threat not “thread”?

2.    Related Works:

It is rarely seen that related work is separated from Introduction. I suggest that the authors should combine it with Introduction. In Introduction part, by analyzing what others have done before in using object detection method, the authors can point out the progress that they made in this study.

3.    Materials and Methods

Line 104: More details should be given, for example, what are the three study areas, what’s the area for each place, when the UAV flights were taken for each place during one year, how many area of interest for each place the authors have identified for training and so on. The objectives here are to give us more background information.

4.    Results and Discussion

Line 171 and 185: The titles for 4.1 and 4.2 are not appropriate. The titles should reflect the real content not just a meaningless description. For 4.1, the title can be ‘Precision results of three CNN-based methods’ or others.

Line 181: No real discussion. Why RetinaNet has the best result compared with the other two methods?

Line 186: The authors have noticed that the tree color, lighting condition, scale and view angle can have effect on the detection result. Besides just listing all the cases, the authors should give us more description. The authors have concluded that all the methods used in this study perform well than other detection approaches (Line 188). What’s the criterion that help the authors drew this conclusion?

Figure 4 has been utilized to show detection results in different seasons (Line 186), but I cannot find the seasonal difference for the 4 subplots in Figure 4 and I can just find the different bounding boxes generated from three different methods.

Figure 5: What is the lighting and scale for each subplot?

Figure 6: What is the view angle for each subplot?

Line 197: This part can be put in discussion part.

My last question is that the trees the authors interested in for this study are growing in groups as we can see from the pictures, so they use bounding box to identify them. If the trees are sparsely planted, will the methods used here still be able to identify the single tree? If they still can, this study can propose a promising perspective. If not, what further improvements should be made?

Author Response

We appreciate the significant contributions to the improvement of the article. The answers to each of the following suggestions are presented below (highlighted in green):

1. Introduction:

Line 17: Threat not “thread”?

Answer: Sorry for this error. We corrected it.

2. Related Works:

It is rarely seen that related work is separated from Introduction. I suggest that the authors should combine it with Introduction. In Introduction part, by analyzing what others have done before in using object detection method, the authors can point out the progress that they made in this study.

Answer: Implemented. Thank you for the suggestion. We combined both sections and addressed the novel contribution in the introduction section, as suggested.

3. Materials and Methods

Line 104: More details should be given, for example, what are the three study areas, what’s the area for each place, when the UAV flights were taken for each place during one year, how many area of interest for each place the authors have identified for training and so on. The objectives here are to give us more background information.

Answer: Implemented. More details regarding the study areas were added such as the number of trees and the total area of the study regions. We also included a mention that the regions are located in the urban part of Campo Grande municipality.   

4. Results and Discussion

Line 171 and 185: The titles for 4.1 and 4.2 are not appropriate. The titles should reflect the real content not just a meaningless description. For 4.1, the title can be ‘Precision results of three CNN-based methods’ or others.

Answer: Implemented. We changed the section titles as follows, 3.1 ‘Precision results of three CNN-based methods’ and 3.2 'Detection under Different Conditions' as suggested. 

Line 181: No real discussion. Why RetinaNet has the best result compared with the other two methods?

Answer: Implemented. To address this issue, we included the following discussion (lines 171-173).

RetinaNet proposed a new loss function to focus learning on hard negative examples. In this case, training focused on separating the Cumbaru from similar trees (hard examples) contributed to greater precision.

Line 186: The authors have noticed that the tree color, lighting condition, scale and view angle can have effect on the detection result. Besides just listing all the cases, the authors should give us more description. The authors have concluded that all the methods used in this study perform well than other detection approaches (Line 188). What’s the criterion that help the authors drew this conclusion?

Answer: Implemented. We included the discussion below in lines 181-187 to address this issue. 

In contrast to other detection approaches (e.g., [12]) all tested methods managed to perform well regardless of image acquisition dates. Previous work suggested periods of the year best suited for capturing images (e.g., September [38] and October [39]). Voss and Sugumaran [12] showed that methods trained in images captured in the fall present more consistent results to those trained with images captured in the summer. On the other hand, the methods used in this work do not need to be trained separately in each season and present consistent precision compared to the literature methods.

Figure 4 has been utilized to show detection results in different seasons (Line 186), but I cannot find the seasonal difference for the 4 subplots in Figure 4 and I can just find the different bounding boxes generated from three different methods.

Answer: The four subplots in each row correspond to the detection results of the four CNNs. The two rows present images in different seasons in which we can perceive the different appearance of the trees. The first row of images shows the Cumbaru with chestnuts while the second row presents the Cumbaru with greener leaves. We included this discussion in lines 178-181 to improve the reading of the manuscript.

Figure 5: What is the lighting and scale for each subplot?

Answer: The scale and lighting were estimated, and now is presented in the figure caption. 

Figure 5. Examples of detection results in images captured for different lighting (average of 67.15 and 130.99 for the brightness channel of the HSB color space) and scale (1:4,000 and 1:2,500) conditions. Columns correspond from left to right to the (a) ground truth, (b) Faster-RCNN, (c) YOLOv3, (d) RetinaNet.  

Figure 6: What is the view angle for each subplot?

Answer: The angles for each subplot are presented in the figure caption now, as required.

Figure 6. Examples of detection results in images with different capture angles (0º and 30º). Columns correspond from left to right to the (a) ground truth, (b) Faster-RCNN, (c) YOLOv3, (d) RetinaNet.  

Line 197: This part can be put in discussion part.

Answer: We changed the title section to "Discussion on Computational Complexity".

My last question is that the trees the authors interested in for this study are growing in groups as we can see from the pictures, so they use bounding box to identify them. If the trees are sparsely planted, will the methods used here still be able to identify the single tree? If they still can, this study can propose a promising perspective. If not, what further improvements should be made?

Answer: The applied technique based on CNN and UAV RGB imagery enables the detection of individual trees. The images presented in Figures 2, 4, 5 and 6 are from individual trees. In the title, we used the term “individual” to highlight this characteristic

Reviewer 2 Report

The paper is competently presented and clear. It is not exactly novel, but worth reading and the authors do not belabor the results. More care is needed in the writing and citing of code-bases used to heighten the reproducibility of the work and increase the value to the community.

Comments

However, to the best of our knowledge, 31 no study focused thus far on state-of-the-art deep learning-based methods for tree detection on images 32 generated by RGB cameras on board of UAVs”

See recent work (not UAV): https://www.mdpi.com/2072-4292/11/11/1309

In general, I would avoid the “we are the first” sentiment. Focus rather on what the paper offers, not some grand statement about what hasn’t been done.

L200: Assuming the authors did not reimplement the CNNs, please cite the repos or codebases you used for the Yolo, RCNN and retinanet. The maintainers of these tools deserve credit.

Minor Comments

The article has many typos and several misspellings. The written English is fine, but there are a lot of places where it lacks attention to detail (i.e, there are 13 authors, but no one caught a typo in the very first sentence?) L17: “Thread” -> “Threat”. There is just some text in blue on L81?) This kind of thing makes the reader (and reviewer) doubt whether the work was done with care.

This is a minor comment, but I’m concerned about the number of epochs in the retinanet training. I’ve used that model extensively and never trained for more than 40 epochs for 10,000 images. Can the authors place some of the training curves in the sup materials? I am shocked that this doesn’t overfit. The cross-fold strategy must have fairly similar images.  

Author Response

We appreciate the significant contributions to the improvement of the article. The answers to each of the following suggestions are presented below (highlighted in green):

The paper is competently presented and clear. It is not exactly novel, but worth reading and the authors do not belabor the results. More care is needed in the writing and citing of code-bases used to heighten the reproducibility of the work and increase the value to the community.

Answer: Thank you for your careful analysis. We performed all the suggested corrections.

Comments

“However, to the best of our knowledge, 31 no study focused thus far on state-of-the-art deep learning-based methods for tree detection on images 32 generated by RGB cameras on board of UAVs” See recent work (not UAV): https://www.mdpi.com/2072-4292/11/11/1309

Answer: We included the reference in the introduction section.

In general, I would avoid the “we are the first” sentiment. Focus rather on what the paper offers, not some grand statement about what hasn’t been done.

Answer: We reformulated the phrase focusing on what we are proposing.

L200: Assuming the authors did not reimplement the CNNs, please cite the repos or codebases you used for the Yolo, RCNN and retinanet. The maintainers of these tools deserve credit.

Answer: We included the repos used as the basis for our implementation in the manuscript (line 100 and footnote 3). 

Minor Comments

The article has many typos and several misspellings. The written English is fine, but there are a lot of places where it lacks attention to detail (i.e, there are 13 authors, but no one caught a typo in the very first sentence?) L17: “Thread” -> “Threat”. There is just some text in blue on L81?) This kind of thing makes the reader (and reviewer) doubt whether the work was done with care.

Answer: We apologize for these errors, which have been corrected in the text. A detailed revision was performed in the text.

This is a minor comment, but I’m concerned about the number of epochs in the retinanet training. I’ve used that model extensively and never trained for more than 40 epochs for 10,000 images. Can the authors place some of the training curves in the sup materials? I am shocked that this doesn’t overfit. The cross-fold strategy must have fairly similar images. 

Answer: RetinaNet did not overfit during training using our image dataset as can be seen in the figure below. This figure shows the loss function in the training and validation sets.

Round 2

Reviewer 1 Report

The authors have answered my questions and I think it is suitable for publishing.

Reviewer 2 Report

As I said in the first review, the paper is interesting and worthwhile. It is not particularly novel, but these kind of benchmark comparisons are useful in saving future researchers time. I appreciate the work and found it helpful. Thank you for a clear revision.